# Effects of Silage Type and Feeding Intensity on Carcass Traits and Meat Quality of Finishing Holstein–Friesian Bulls

**DOI:** 10.3390/ani13193065

**Published:** 2023-09-29

**Authors:** Zenon Nogalski, Monika Modzelewska-Kapituła, Katarzyna Tkacz

**Affiliations:** 1Department of Animal Nutrition, Feed Science and Cattle Breeding, Faculty of Animal Bioengineering, University of Warmia and Mazury in Olsztyn, Oczapowskiego 5, 10-719 Olsztyn, Poland; zena@uwm.edu.pl; 2Department of Food Microbiology, Meat Technology and Chemistry, Faculty of Food Sciences, University of Warmia and Mazury in Olsztyn, Plac Cieszyński 1, 10-719 Olsztyn, Poland; ktkacz@uwm.edu.pl

**Keywords:** beef, diet composition, feed conversion ratio, shear force, sensory evaluation

## Abstract

**Simple Summary:**

In many countries, dairy bulls in beef production are used. The main flaw in their rearing is the difficulty in obtaining the optimal intramuscular fat content, affecting the quality properties consumers appreciate. In the present study, a 4-month finishing phase was applied to improve the meat quality of Holstein–Friesian bulls. Two levels of energy concentration in the feed portion and two types of silages were used. The silage type did not affect the carcass or meat quality, whereas the concentration of energy did. The intensive feeding beneficially affected performance, carcass value and the quality of Holstein–Friesian bulls’ meat.

**Abstract:**

The study aimed at evaluating the influence of silage type (grass silage—GS, and maize silage—MS) and energy level in the feed portion on animals’ performance, carcass value and the quality of Holstein–Friesian bull meat. The animals were reared using intensive (I) feeding, 1.02 feed units for meat production (UFV)/kg dry matter (DM), and semi-intensive (SI) feeding, 0.94 UFV/kg DM. Thirty-two HF bulls with an average live weight of 530 kg were assigned to four feeding groups. The proportions (g/kg DM) of feed in the diets were as follows: group GS-I, GS 500, concentrate 500; group GS-SI, GS 700, concentrate 300; group MS-I, MS 500, concentrate 500; and group MS-SI, MS 700, concentrate 300. After 4 months of the finishing phase, bulls were slaughtered and samples of the *longissimus lumborum* muscle were collected. The silage type did not affect performance, carcass value or meat quality. However, in the intensive feeding, a 33% increase (*p* < 0.01) in daily weight gain and a reduction in the feed conversion ratio compared to semi-intensive feeding were noted. The carcasses of bulls fed intensively received higher scores for conformation and fatness than the carcasses of bulls fed semi-intensively. The meat of I group bulls had a higher intramuscular fat content and received higher juiciness, tenderness, taste and overall acceptability scores.

## 1. Introduction

Intramuscular fat (IMF) or marbling is a meat attribute appreciated by consumers because of its positive impact on meat’s tenderness, taste and juiciness [1]. Changes in IMF content are associated with modifications in the flavour desirability of the beef because of its impact on the fatty acid composition and volatile compounds produced in beef during cooking [2]. The main factors that influence the IMF content in beef are breed, sex, live weight at slaughter and feeding intensity [3]. Breeds such as Wagyu and Hanwoo show a high-marbling potential (up to 38% and 20% IMF in *longissimus* muscle, respectively), whereas Holstein–Friesian (HF), Simmental, Nellore and Belgium Blue have much lower marbling (up to 5% of IMF) [4]. The beef originating from bulls contains less IMF than from heifers. Moreover, castration, early weaning and high-energy feeding increase the IMF content, as well as higher slaughter age and live weight [4]. In countries where the proportion of beef breeds is low, beef is produced using dairy breeds. Milk producers get rid of male cattle offspring, which makes them available for beef producers and therefore beef is mainly produced from bulls. This is the case in Poland, where more than 50% of beef is obtained from young bulls, mostly belonging to the HF breed [5]. It should be noted that the HF breed is characterized by large and late maturing animals whose carcass quality improves along with an increase in live weight [6]. In contrast to small and early maturing cattle breeds, HF is suitable for long fattening, which does not lead to excessive subcutaneous fat tissue deposition or deterioration of the carcass quality [7]. Therefore, heavy fattening of HF bulls is recommended until a live weight of 700–800 kg [1]. Due to the high cost of concentrate, the most popular is a semi-intensive feeding system, in which silage feeding *ad libitum* is used with a small proportion (1–2 kg) of concentrate [8]. However, feeding roughage with a small addition of concentrates increases the fattening time, increasing the age at slaughter, which may negatively influence the sensory characteristics of beef, especially its tenderness [9]. An alternative might be a feeding method in which there is a finishing phase when an energy-dense diet, typically having increased concentrate proportion in the diet, is used. The application of the finishing phase in bull feeding can increase the carcass weight and meat quality due to an increase in IMF [10]. The use of high-concentrate diets at the finishing phase can also shorten the animal production cycle [11]. In Poland, 90% of beef is produced in dairy herds and, therefore, the application of the optimal Polish HF bull feeding method with the finishing phase might improve beef quality [5].

In a temperate climate, beef production is based mainly on maize and grass. The feeding using maize silage (MS) is more intensive compared to grazing or using grass silage (GS). When using MS, higher daily gains and shorter fattening periods are reported [12]. MS contains more starch and has higher digestibility than green grass and GS [13]. However, when using forage and GS instead of MS, it is possible to improve protein supply in a basal diet, which in consequence reduces the protein demand from other additional sources [14]. Another added value of using GS in cattle feeding is obtaining beef with a higher n-3 polyunsaturated fatty acid concentration and a more beneficial, lower n-6/n-3 ratio [15], which is desirable from a human nutrition perspective [16]. However, the changes might result in modification of the sensory quality related to the presence of fish or grass aroma, which might decrease consumer acceptance [17]. Also, feeding intensity is a key factor affecting animals’ performance and carcass value [18]. Feeding intensity that is too high generates excessive costs, whereas too low is not sufficient to obtain a desirable IMF in muscles which negatively affects the sensory quality of beef [19]. Therefore, the study aimed to evaluate the impact of silage type and feeding intensity during the finishing phase on performance, carcass value and meat quality of young HF bulls. A hypothesis that MS and the supplementation of a high proportion of concentrate has a more beneficial impact on finish progress and meat quality than the GS and the supplementation of a lower proportion of concentrate was tested.

## 2. Materials and Methods

### 2.1. Animals and Feeding

The experiment comprised 32 Polish HF bulls reared at the Agricultural Experiment Station in Bałcyny (53°35′29″ N; 19°50′58″ E; Poland). The climate of the region is described as temperate marine continental. The average length of the growing season is 190 days (15 April–25 October). Calves were reared in a traditional system, fed using milk replacer, hay and concentrate. When young bulls were 5 months old, the semi-intensive feeding began. They were kept in group pens and fed *ad libitum* total mixed ration (TMR) composed of GS and MS supplemented with 2 kg of concentrate.

At the age of approx. 600 days, when bulls reached a live weight (LW) of approx. 530 kg, they were assigned to four groups by the analogue method (the animals were at a similar age, and they were divided into groups based on LW) of eight individuals each. They were placed in a monitored feeding facility in four separate pens on deep bedding. The pen size was 20 × 13 m (8.12 m^2^/bull). The animals had free access to water and salt licks. For a two-week pre-experimental period, the bulls were adapted to housing conditions and feeding. After this, the feeding experiment started. During a finishing phase, which lasted 4 months, the animals were fed with two different types of silage and supplemented with two different levels of concentrate (Table 1). To produce GS, first-harvest herbage cut in the heading stage and then wilted for 24 h was used, whereas MS was obtained from herbage cut in the dough stage (Table 1). GS was produced from a mix of Milkway universal (Barenbrug Sp. z o.o., Tarnowo Podgórne, Poland) composed of perennial ryegrass (40%), multiflower ryegrass (10%), meadow fescue (30%) and meadow timothy (20%). MS was produced from Goldnugget maize. Triticale grain of the Dinaro variety harvested in 2022 was used. The grain was ground before feeding.

The dietary treatments were as follows in the proportion (g/kg dry matter): group GS-I (Grass silage-Intensive feeding), GS 500, concentrate 500; group GS-SI (Grass silage-Semi-Intensive feeding), GS 700, concentrate 300; group MS-I (Maize silage-Intensive feeding), MS 500, concentrate 500; and group MS-SI (Maize silage-Semi-Intensive feeding), MS 700, concentrate 300 (Table 2). The crude protein concentration was constant for all groups, whereas the energy level differed: for intensive feeding (GS-I and MS-I groups), it was 1.02 UFV/kg DM, whereas for semi-intensive feeding (GS-SI and MS-SI), it was 0.94 UFV/kg DM. TMR composed of silage, concentrate and the commercial mineral–vitamin premix (2% of DM) was offered to animals *ad libitum*. TMR was dosed from a self-propelled feed cart. Commercial mineral–vitamin premix for cattle (code of product 7619; Cargill Poland Ltd., Warsaw, Poland) consisting of, per kg: Ca, 235 g; Na, 79 g; P, 48 g; Mg, 28 g; Fe, 500 g; Mn, 2000 mg; Cu, 375 mg; Zn, 3750 mg; J, 50 mg; Co, 12.5 mg; Se, 12.50 mg; vitamin A, 250,000 IU; vitamin D3, 50,000 IU; vitamin E, 1000 mg; and dl-alpha-tocopherol, 909.10 mg was used. The animals were weighed at the beginning of the experiment and then once a week. The bulls received fresh TMR every day when leftovers were removed. The Roughage Intake Control System (Insentec BV, Marknesse, The Netherlands) was used to monitor the individual daily feed intake of each animal. Feed samples were collected once a week to assess the content of basic nutrients [20], as well as NDF, ADF and ADL [21]. The value of pH in silage samples was measured with HI 8314 pH-meter (Hanna Instruments Polska, Olsztyn, Poland). Moreover, silage sample contents of lactic acid, acetic acid and butyric acid were determined in water extract using high-performance liquid chromatography (the Shimadzu system equipped with a Varian Meta Carb 67H column 67H P/N 5244 column (Varian, Palo Alto, CA, USA). To determine the content of water-soluble carbohydrates in silage, the anthrone method [22] was used. The content of protein nitrogen (N-protein) was analysed with the use of trichloroacetic acid (TCA), whereas ammonia nitrogen (N-NH_3_) was analysed according to the Conway method [23]. Amino acid nitrogen was calculated based on the number of free amino acids, which were determined with the AAA 400 INGOS automatic amino acid analyser (Czech Republic) with the use of a lithium column after deproteinization of TCA samples. The animals remained under veterinary supervision. No health problems were noted in any of the animals during the experiment.

Finishing fattening was carried out for 120 days. Weight gains during the finishing phase were calculated by subtracting the initial weight from the final weight, and the resulting difference was divided by 120.

### 2.2. Carcass Quality

At the end of the finishing phase, the bulls were transported to a slaughterhouse where they rested for the next 15 to 20 h in individual pens with access to water. Before slaughter, animals were weighed with an accuracy of 0.5 kg and the live weight (live weight before slaughter) was recorded. The process of slaughter and postslaughter operations were carried out in accordance with Council Regulation (EC) No. 1099/2009 of 24 September 2009 [24]. After postslaughter processing, carcasses were weighed with an accuracy of 0.5 kg (hot carcass weight). Based on the live weight before slaughter and carcass weight, the dressing percentage was calculated in relation to the live weight before slaughter. Carcass conformation and fat cover were assessed by a professional grader in the EUROP system [25]. The value of meat pH_48h_ was measured after 48 h of carcass chilling, in the *longissimus thoracis* muscle, between the 10th and 11th thoracic vertebrae with the use of a Double Pore combination electrode (Hamilton Bonaduz AG, Bonaduz, Switzerland) and a pH 340i pH-meter with a TFK 150/E temperature sensor (WTW, Xylem Inc., Washington, DC, USA).

### 2.3. Meat Sampling

For meat quality analyses, *longissimus lumborum* (LL, n = 32) was removed from the left half-carcasses (between the 1st and 6th lumbar vertebrae) 96 h postmortem. Transport of samples in a portable refrigerator to a laboratory lasted about 1 h. The LL samples were kept at the temperature of 4 °C (±1 °C) overnight. The next day, the samples (approx. 700 g) were individually vacuum packaged in PA/PE (thickness 70 µm; Inter Arma Ltd., Rudawa, Poland) and placed into a climatic chamber [Memmert GmbH, Schwabach, Germany] at 4 ± 1 °C until day 14 postmortem [26]. After that, the LL samples were put in the freezer and stored at −20 °C until analyses (approx. 4 months). Before analyses, samples were thawed at 4 ± 1 °C for 48 h. The colour, chemical composition (moisture, protein, fat, and ash contents), pH value and free water content were determined in 300 g samples obtained from each LL. From the remaining meat sample, two approx. 200 g 2.54 cm thick steaks were cut. After weighing, the steaks were placed in individual plastic bags, put into a water bath (80 °C, Aquarius M/150Z, Aqua Lab, Warsaw, Poland) and heated to 71 °C inside the steaks. The temperature inside a steak was monitored continuously with an electronic thermometer. After thermal treatment, one steak was used to determine sensory quality and cooking loss and the other to determine Warner–Bratzler shear force (WBSF) values.

### 2.4. Chemical Composition

For the chemical composition analyses (moisture, protein, fat, and ash contents), approx. 200 g of LL after 14 d ageing was ground twice using 6 mm and 3 mm meshes and then thoroughly manually mixed. Samples were analysed using an Association of Official Analytical Chemists approved [27] near-infrared spectrophotometer (FOSS FoodScanTM 2 Lab/Pro; FOSS Analytical A/S, Hillerød, Denmark). The device operates according to the standard PN-A-82109:2010 [28]. Independent readings (n = 16) were taken from each sample and averaged to obtain the final reported values.

### 2.5. pH Value

The pH values of LL were measured in triplicate after thawing directly in meat using an FC 200 combination electrode connected to a HI 8314 pH-meter (Hanna Instruments Polska, Olsztyn, Poland) and mean values for each group were reported. The device was calibrated using pH 7 and pH 4 buffers before the measurements.

### 2.6. Free Water

The free water content was determined by the Grau and Hamm method [29] with the use of computer image analysis [30]. Briefly, 0.3 g samples of ground meat were placed on a filter paper (Whatman 1, Whatman Laboratory Division, Maidstone, UK) between two glass tiles. A weight of 2 kg was applied to each sample for 5 min. Then, the paper surface with pressed meat sample and liquid areas was photographed with a digital camera (Nikon D90, Nikon Corporation, Tokyo, Japan). Measurements of stain areas were taken using Nikon NIS-Elements BR 2.20. software (Nikon Corporation, Tokyo, Japan). Free water (FW) content was calculated based on areas of pressed meat samples, liquid area and filter paper absorbability (Equation (1)).
(1)$$FW=\frac{a\cdot (l-m)}{c}\cdots (\%)$$
where:

*a*—water absorbability of filter paper (cm^3^), *l*—an area of liquid (cm^2^), *m*—an area of pressed meat sample (cm^2^), *c*—a mass of sample (g). The analysis was performed in triplicate for each LL.

### 2.7. Meat Colour

Meat colour was measured in the CIELab system [31] from a freshly cut surface after 25 min blooming [32] using a Konica Minolta CR-400 (Sensing Inc., Osaka, Japan, 2° view angle, D65 illuminant) at three randomly selected points. The values of lightness (L*), redness (*a**), and yellowness (*b**) were measured, and mean values for each group were reported. Based on *a** and *b** values, Chroma (*C*) and hue (h) were calculated according to Equations (2) and (3).
(2)$$C=\sqrt{\left(a^{*}{}^{2}+b^{*}{}^{2}\right)}$$
(3)$$H=\mathrm{atan}(\frac{b^{*}}{a^{*}})\cdot 180/PI$$

### 2.8. Cooking Loss and Warner–Bratzler Shear Force Values

Cooking loss was calculated as the difference in sample weight before and after thermal treatment, and it was expressed as % of the initial sample weight [33].

To determine Warner–Bratzler Shear Force (WBSF, N) values, steaks heated in a water bath at 80 °C to obtain 71 °C inside were used. After termination of heating, the steaks were chilled overnight (at 4 °C) and samples (n = 5 from each steak, 10 × 10 mm) were cut. The samples were cut from steaks parallel to the longitudinal orientation of muscle fibres. WBSF values were determined using the Instron 5942 universal testing machine (Instron, Norwood, MA, USA) equipped with a V-shaped shear blade with a triangular aperture of 60°. The samples were cut (load 500 N, head speed 200 mm/min) perpendicularly to the longitudinal orientation of the muscle fibres at room temperature (approx. 20 °C). Mean values calculated from 5 measurements were reported.

### 2.9. Sensory Assessment

The sensory quality of the beef was determined immediately after the termination of the thermal treatment, according to Standard PN-ISO 4121 [34]. The evaluation was performed by a six-person team trained and experienced in sensory analyses of meat [35]. Warm samples were cut into approximately 2 mm thick slices, coded with three-digit numbers, and presented to the panellists on white plates in random order. The evaluation was carried out at a room temperature of approximately 20 °C, under white fluorescent lighting. Water and bread were provided for cleansing the palate. A total of eight sensory analysis sessions were performed. During each session, four meat samples were assessed. The same panellists participated in all evaluation sessions. The panellists evaluated each sample using a scale from 1 to 10 in terms of meat aroma intensity (1, imperceptible; 10, extremely intense), juiciness (1, extremely dry; 10, extremely juicy), tenderness (1, extremely tough; 10, extremely tender), meat taste intensity (1, imperceptible; 10, extremely intense) and overall acceptability (1, unacceptable I extremely dislike; 10, excellent, I like it very much).

### 2.10. Statistical Analysis

A statistical analysis of the results was conducted using Statistica 13 [36] software. The influence of silage type (GS and MS) and feeding intensity (I and SI) on animals’ performance, carcass value and meat quality attributes were determined by the least squares method, using the formula:*Y_ijk_* = μ + *A_i_* + *B_j_* + (*AB*)*_ij_* + *e_ijk_*
where: *Y_ijk_* is the value of the analysed parameter, μ is the population mean, *A_i_* is the effect of silage type (1, 2), *B_j_* is the effect of feeding intensity (1, 2), (*AB*)*_ij_* is silage type × feeding intensity interaction, and *e_ijk_* is the random error. To analyse the results of sensory analysis, nonparametric Scheirer–Ray–Hare tests were applied to evaluate the effects of two factors and compare several averages [37]. The significance level was set at *p* < 0.05 and at *p* < 0.01.

## 3. Results

### 3.1. Feeding Results

The animals fed intensively obtained 33% higher (*p* < 0.01) daily weight gains compared with those fed semi-intensively and, as a consequence, a higher live weight at the end of the finishing phase was noted (Table 3). Silage type did not affect growth rate. There was an interaction between the factors for daily body weight gains (Figure 1). MS in the intensive feeding generated a higher weight gain than GS; however, no such differences were noted in semi-intensive feeding. The feed intake of the bulls in intensive diet was higher (*p* < 0.05) and the feed conversion ratio was lower (*p* < 0.05) compared to the semi-intensive diet.

### 3.2. Carcass Quality

Live weight before slaughter, similar to that at the end of the finishing phase, was higher in bulls from intensive feeding compared to bulls from semi-intensive feeding (*p* < 0.001, Table 4). The carcasses of the intensively fed animals were heavier by about 39.6 kg, which accounted for 11% compared to semi-intensively fed bulls. Dressing percentage was higher (*p* < 0.05) in intensively fed bulls compared to semi-intensively fed bulls. Conformation and fat scores of intensively fed bulls were higher (*p* < 0.01) than semi-intensively fed bulls.

### 3.3. Chemical Composition and Physicochemical Properties

Feeding intensity significantly affected IMF and protein content (Table 5). In the LL samples of intensively fed bulls, the concentration of fat was higher (*p* < 0.01) and the concentration of protein lower (*p* < 0.05) compared to semi-intensively fed animals. No significant differences in the chemical composition between groups receiving different silage types were noted. There was no interaction between silage type and feeding intensity (Table 5). No significant effect of silage type nor feeding intensity on the physicochemical properties of the meat was noted (Table 5).

### 3.4. Sensory Quality

The results of the sensory assessment were shown in Table 6. Feeding intensity affected significantly all sensory attributes, except for meat aroma intensity. Juiciness, tenderness, taste and overall acceptability of meat samples from intensive feeding were scored higher than those from the semi-intensive system. However, no significant effect of silage type on the sensory quality was noted, nor interaction between silage type and feeding intensity (*p* > 0.05).

## 4. Discussion

The rate of daily weight gain is a direct indicator of the evaluation of the growth potential of HF bulls. In the study, when comparing intensively fed groups (50% DM from concentrate) with semi-intensively fed groups (30% DM from concentrate), differences toward a more beneficial intensive system were noted—a higher feed intake, and daily weight gains and a lower FCR. It was a consequence of a lower NDF content and higher energy of the TMR fed in intensive finishing. FCR represents the amount of feed intake divided by live weight gain and indicates improvements in feed conversion [38]. Higher live weight and daily gain of cattle as a result of the increased energy value of feed were reported also by Therkildsen et al. [39]. The results of the study by Huuskonen and Huhtanen [40] showed that in cattle feeding, energy intake had the greatest impact on daily body weight gain, while the protein content in the diet had little impact on the growth rate. Results of studies conducted by Jennings et al. [41] suggested that feeding excess crude protein and metabolizable protein will increase maintenance energy requirements of finishing steers.

In the present study, MS did not prove to be as indicated by other authors [12,13,15], in which there was higher suitability for intensive cattle feeding compared with GS. According to Reynolds et al. [42], differences in NDF concentration between feeds influence the intensity of daily gains. Feeds containing more NDF result in greater energy use for digestion, especially chewing and movement of food.

In the present study, intensively fed young bulls showed better slaughter value scores. In the previous studies on young bulls that were beef and dairy crossbreds [7] and steers [8], intensive feeding increased the dressing percentage. It resulted from a higher fat deposition (fatness score) and better carcass conformation, which was also noted in the present study. Carcass conformation increases along with live weight before slaughter and feeding intensity [7]. It should be highlighted that feed costs constitute the main part of the total costs of beef production, regardless of the fattening system adopted [41]. In our research, the intensive system was associated with greater consumption of concentrate feed. Therefore, the effect of increased slaughter value obtained in the intensive system should be compared with the increased costs of feeding these bulls.

The values of pH measured at 48 h post-slaughter and after 14 d ageing and frozen storage were typical for normal-quality meat [43]. This indicates that animals were handled properly before slaughter and there was a lack of DFD (dark, firm dry) defects, which would decrease beef quality and production profitability [44]. The pH value is a key factor affecting the technological quality of meat, including its shelf-life, water holding capacity and even sensory quality attributes [45]. The lack of differences between treatments in the present study in pH values corresponds with no differences in water holding capacity (free water contents, cooking loss), and meat colour. The results of the present study stay in agreement with the findings of other authors. No effect of feeding intensity on LL pH was noted by Sami et al. [46] in Simmental bulls. Also, Keller et al. [47], who used different diet compositions, including grass and MS, in Limousin × dairy crossbred bull feeding reported no effect on *longissimus thoracis et lumborum* (LTL) pH.

In the present study, a significant increase in fat content as a result of an intensive feeding system was noted. It should be highlighted that the average fat content in the meat of intensively fed bulls was 3.5%, whereas in semi-intensively fed it was 2.25%. A product that contains less than 3% of fat is regarded as low fat according to European Union Regulation (EC) No 1924/2006 [48]. Therefore, only the meat from semi-intensively fed bulls meets the criteria for labelling as “low fat”. The increase in the fat content as a result of intensive feeding beneficially affected sensory attributes of the meat, which becomes more juicy, tender, and tastier and was scored by panellists as more acceptable than the meat from semi-intensive feeding. These findings highlight the role of fat in shaping the sensory profile of meat and meat products. The increase in the IMF content along with increased feeding intensity was also noted in our previous studies [1,19,49]. Fat content in the *infraspinatus* muscle increased from 3.74% in the semi-intensive feeding system to 4.11% in intensive feeding, although with no effect on the sensory quality or WBSF values [49]. However, in young crossbred (HF × Hereford) bulls from intensive feeding, the increase in fat content in LL muscles (from approx. 1.8% to 3.0%) was accompanied by a reduction in WBSF and improved sensory quality [1]. This leads to the conclusion that different muscles of beef carcasses might differently respond to the increased feeding intensity. It should be also highlighted that attributes such as texture, colour and sensory quality are affected by many different factors apart from IMF content (e.g., cattle breed, age, post-mortem carcass handling, meat ageing etc.). In the present study, the increase in the IMF content was high enough to improve sensory quality; however, it did not cause changes in colour or instrumentally measured texture of the meat. A similar effect was noted by Sami et al. [46] in intensively reared Simmental bulls. On the other hand, Honig et al. [50] reported that feeding German Simmental bulls high-energy rations did not lead to changes in the chemical composition of LTL. Similar to the findings of the present study, Honig et al. [50] also reported that feeding intensity did not affect cooking loss, WBSF, pH or meat colour. No effect of diet composition, i.e., the proportion of grass and MS on the water-holding ability in meat (ageing loss, drip loss, cooking loss) was noted by Keller et al. [47], which supports the results of the present study.

In the present study, no effect of feeding intensity or silage type on LL colour was shown. Similar results were reported by Keller et al. [47], Keady et al. [51] and Keady et al. [13], who did not show significant differences in LTL colour as a result of diet modification by replacing MS with GS. The colour attributes values of raw beef used in the present study resembled those noted in previous studies for LL obtained from HF carcasses [52]. The colour of fresh beef is one of the attributes based on which consumers make their decision about the purchase [53]. The most acceptable by consumers is a bright, cherry-red beef, with a* values equal to or higher than 14.5, L* higher than 31.4 and b* higher than 6.3, respectively [54]. These thresholds were achieved among all treatments in the present study, which indicates that beef from all experimental groups would be attractive visually for consumers. Due to the lack of differences in colour between groups receiving different silages and feeding intensity, it might be concluded that consumers will not differentiate the meat obtained using feeding strategies investigated in the present study at the stage of purchase.

In the present study, no significant effect on WBSF was noted. These results corroborate with findings of Keady et al. [51], who did not note the effect of GS replacement by MS on WBSF. In the present study, meat tenderness was evaluated in instrumental analysis (WBSF) and sensory analysis. The beef from the intensive system was rated higher in sensory assessment. However, treatments did not differ in shear force values, which presents the force needed to bite meat during consumption. It should be highlighted that all treatments showed low WBSF values, which were below 34 N. In the literature, different thresholds for tender beef are proposed, such as 41–49 N [47], or a more detailed classification, including very tender meat (up to 32.96 N), tender (up to 42.77 N), and acceptably tender (up to 52.68 N) [55]. The meat obtained from all treatments in the present study therefore might be classified as tender, but these from the MS-I group were classified as very tender.

## 5. Conclusions

In the present study, the advantage of the increased energy level and reduced concentration of fibre in the feed dose in the finishing phase of HF young bulls was shown and manifested in higher weight gains, feed intake and feed conversion ratio. Intensive feeding increased the hot carcass weight, conformation score and fatness score, as well as the IMF content in LL and improve its sensory properties. No significant impact of silage type on carcass performance and meat quality was noted. It should be highlighted that the intensive feeding enabled us to produce good quality meat from Polish HF bulls.

The hypothesis that the application of MS and the supplementation with a high proportion of concentrate have a beneficial impact on animals’ performance was proved only in terms of a high proportion of concentrate in a feed ration. However, further investigation to assess the effect of silage type on the nutritional quality of meat is needed.

## Figures and Tables

**Figure 1 animals-13-03065-f001:**
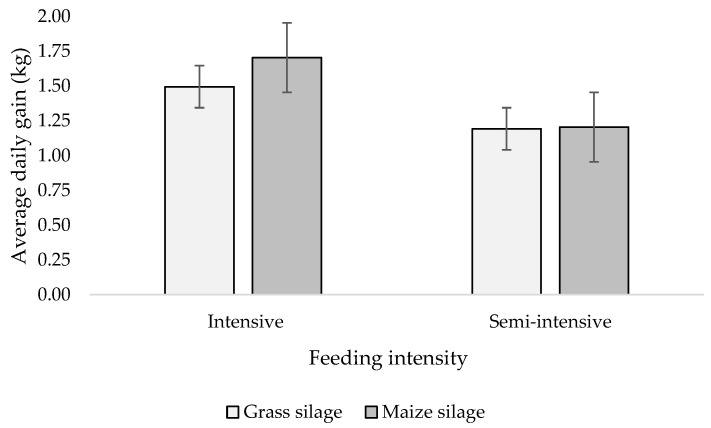
Feeding intensity × silage type interaction on average daily gain (vertical bars refer to the standard error of the mean, SEM).

**Table 1 animals-13-03065-t001:** Chemical composition of experimental feed.

Specification	Grass Silage	Maize Silage	Triticale	Rapeseed Meal
Dry matter (DM, g/kg)	285	322	875	878
Organic matter (g/kg DM)	906	967	966	921
Crude protein (g/kg DM)	121	88.9	122	383
NDF (g/kg DM)	536	337	162	298
ADF (g/kg DM)	314	196	41	212
ADL (g/kg DM)	25.5	12.7		
NFC (g/kg DM)	194	508	629	237
pH	4.23	3.56		
Lactic acid (g/kg DM)	43.6	27.5		
Acetic acid (g/kg DM)	12.5	6.6		
Butyric acid (g/kg DM)	0.09	0.08		
N-NH_3_ (g/kg TN)	75.9	33.9		
UFV	0.86	0.85	1.19	1.03
PDIN	83	50	85	254
PDIE	72	68	98	162

NDF—neutral detergent fibre; ADF—acid detergent fibre; ADL—acid detergent lignin; NFC—nonfibre carbohydrate; N-NH_3_—ammonia nitrogen; TN—total nitrogen; UFV—feed units for meat production, PDIN—protein digested in the small intestine depending on rumen degraded protein; PDIE—protein digested in the small intestine depending on rumen fermented organic matter.

**Table 2 animals-13-03065-t002:** Ingredients (% dry matter, DM) and chemical composition of diets.

Specification	GS-I	GS-SI	MS-I	MS-SI
Grass silage	50	70		
Maize silage			50	70
Triticale grain	47	27	41	18
Rapeseed meal	3	3	9	12
Dry matter (DM, g/kg FM)	580	462	598	488
In g/kg DM				
Organic matter	934.6	922.6	962.4	961.3
Crude protein	129.3	129.1	128.9	130.1
NDF	353.1	427.8	261.7	300.8
ADF	182.6	237.2	133.8	170.0
NFC	399.7	312.7	533.2	497.2
UFV	1.02	0.94	1.02	0.94
PDIN	89.1	88.6	82.7	81.7
PDIE	86.9	82.7	87.7	84.6

GS-I—Grass silage Intensive; GS-SI—Grass silage Semi-Intensive; MS-I—Maize silage Intensive; MS-SI—Maize silage Semi-Intensive; NDF—neutral detergent fibre; ADF—acid detergent fibre; NFC—nonfibre carbohydrate; UFV—feed units for meat production; PDIN—protein digested in the small intestine depending on rumen degraded protein; PDIE—protein digested in the small intestine depending on rumen fermented organic matter.

**Table 3 animals-13-03065-t003:** Effects of silage type and feeding intensity on animals’ performance.

Traits	Silage Type (ST)		Feeding Intensity (FI)		SEM	*p*-Value		
	Grass Silage	Maize Silage	Intensive (I)	Semi-Intensive (SI)		ST	FI	ST × FI
Initial age (days)	604.6	595.8	600.3	599.6	7.36	0.660	0.936	0.695
Initial live weight (kg)	538.9	530.3	531.0	536.5	8.79	0.906	0.866	0.333
Final live weight (kg)	699.9	704.5	723.8	680.7	7.97	0.748	0.005	0.068
Body weight gain (kg)	161.2	174.5	191.9	143.8	3.22	0.114	0.000	0.082
Feeding period (days)	120	120	120	120				
Average daily gain (kg)	1.343	1.454	1.599	1.198	0.0500	0.117	0.000	0.044
Dry matter intake (kg)	10.52	11.05	11.20	10.38	0.081	0.256	0.036	0.369
FCR (kg)	7.83	7.60	7.02	8.66	0.068	0.328	0.045	0.653

FCR, feed conversion ratio, the ratio of dry matter intake to average daily gain; SEM, standard error of the mean.

**Table 4 animals-13-03065-t004:** Basic slaughter traits of the bulls.

Traits	Silage Type (ST)		Feeding Intensity (FI)			SEM	*p*-Value	
	Grass Silage	Maize Silage	Intensive (I)	Semi-Intensive (SI)		ST	FI	ST × FI
Age at slaughter (days)	724.6	715.3	720.8	719.1	8.03	0.896	0.962	0.789
LW before slaughter (kg)	642.2	650.0	673.8	618.4	9.13	0.618	0.001	0.160
Hot carcass weight (kg)	359.4	369.2	385.3	345.7	5.44	0.570	0.002	0.236
Dressing percentage (%)	55.96	56.80	57.18	55.91	0.130	0.603	0.031	0.528
Conformation score (pts)	4.8	4.9	5.2	4.4	0.13	0.597	0.003	0.863
Fatness score (pts)	5.5	5.8	6.4	4.9	0.28	0.538	0.008	0.391

LW, live weight; EUROP conformation was 15 for class E_+_ and 1 for class P_−_; EUROP degree of fat cover was 15 for class 5_+_ and 1 for class 1_−_; SEM: standard error of the mean.

**Table 5 animals-13-03065-t005:** Chemical composition and physicochemical properties of *longissimus lumborum* from bulls-fed grass or maize silage in an intensive or semi-intensive feeding system.

Traits	Silage Type (ST)		Feeding Intensity (FI)		SEM	*p*-Value		
	Grass Silage	Maize Silage	Intensive (I)	Semi-Intensive (SI)		ST	FI	ST × FI
Chemical composition								
Moisture (%)	74.51	73.90	73.80	74.61	0.218	0.152	0.059	0.588
Protein (%)	23.40	23.05	22.97	23.47	0.111	0.092	0.021	0.792
Fat (%)	2.74	3.01	3.49	2.25	0.189	0.412	0.001	0.904
Ash (%)	1.08	1.04	1.04	1.07	0.016	0.253	0.339	0.202
Physico-chemical properties								
pH_48h_	5.65	5.66	5.66	5.65	0.023	0.699	0.719	0.297
pH_14d_	5.64	5.66	5.62	5.68	0.018	0.703	0.084	0.478
Free water (%)	28.34	28.21	26.80	29.74	0.796	0.936	0.073	0.775
L*	35.11	38.36	37.27	36.20	1.530	0.309	0.738	0.797
a*	16.58	15.93	16.52	15.99	0.454	0.483	0.565	0.261
b*	7.66	8.15	8.29	7.52	0.285	0.395	0.179	0.302
C	23.66	20.30	20.73	23.23	1.206	0.166	0.298	0.266
h	24.84	27.46	26.95	25.35	1.109	0.254	0.484	0.846
Cooking loss (%)	34.21	30.54	30.58	34.17	2.295	0.440	0.450	0.443
WBSF (N)	33.94	31.97	32.23	33.69	1.764	0.597	0.694	0.948

WBSF—Warner–Bratzler Shear Force; L*—lightness; a*—redness; b*—yellowness; C—saturation index; h—hue angle; pH_48h_—pH measured 48 h after slaughter; pH_14d_—pH after 14 days of ageing, SEM—standard error of the mean.

**Table 6 animals-13-03065-t006:** Sensory quality of *longissimus lumborum* from bulls fed grass or maize silage in intensive or semi-intensive feeding system.

Traits	Silage Type (ST)		Feeding Intensity (FI)		SEM	*p*-Value		
	Grass Silage	Maize Silage	Intensive (I)	Semi-Intensive (SI)		ST	FI	ST × FI
Aroma	7.1	7.1	7.0	7.2	0.11	0.741	0.487	0.971
Juiciness	5.0	5.1	5.3	4.8	0.15	0.939	0.039	0.564
Tenderness	5.6	5.5	5.8	5.3	0.18	0.914	0.041	0.606
Taste	6.3	6.4	6.6	6.1	0.12	0.819	0.047	0.452
Overall acceptability	6.4	6.2	6.6	5.9	0.13	0.341	0.012	0.861

SEM, standard error of the mean, scale from 1 to 10.

## Data Availability

Datasets generated from the current experiment are available from the corresponding authors upon reasonable request.

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
