# Peer review of "Effects of Silage Type and Feeding Intensity on Carcass Traits and Meat Quality of Finishing Holstein–Friesian Bulls"

_animals, 2023, doi:10.3390/ani13193065_

Round 1

Reviewer 1 Report

This is  a well written study which had the objective of proving the effects of two levels of energy concentration in the fodder portion and two types of silages for fattening Holstein Bulls in Poland.  I have no comments on the soundness of the manuscript, and it is publishable in its present form. However I feel that there is not much novelty in it, because much research has been undertaken in this matter and there are many earlier papers showing the effects of high vs low concentrate on the growth rate and meat quality attributes of bulls with similar results. Perhaps the situation is  different for the country of origin of the study, and then the authors should say so. Moreover, if they recommend the intensive feeding, it would be also valuable to add information regarding the higher costs, because the bulls also had a higher intake of the intensive diet….not only show the cientific results, but also economic results for the producers (the authors show advantages from the consumer perspective but not higher costs for the producers). Please include some comments on this in the discussion.

Specific comments:

L 49-50….large and late maturating animals whose carcass quality improves along with body weight increase [6]. In contrast to small and early maturating cattle…... I am not sure if the correct word is late maturating or late maturing breeds, please check English wording

L169…were used to fabricate 2.54 cm-thick steaks. I suggest to use cut instead of fabricate

Table 5…the SEM of cooking loss (%), 2.295 should be reduced to 2 decimals, as the rest of values

Quality of English language is fine, besides my minor comments

Author Response

Dear Reviewer, thank you very much for your comments. Below we enclosed responses for each comment. In the manuscript, all changes were marked with red font.

C1: This is  a well written study which had the objective of proving the effects of two levels of energy concentration in the fodder portion and two types of silages for fattening Holstein Bulls in Poland.  I have no comments on the soundness of the manuscript, and it is publishable in its present form. However I feel that there is not much novelty in it, because much research has been undertaken in this matter and there are many earlier papers showing the effects of high vs low concentrate on the growth rate and meat quality attributes of bulls with similar results. Perhaps the situation is  different for the country of origin of the study, and then the authors should say so. Moreover, if they recommend the intensive feeding, it would be also valuable to add information regarding the higher costs, because the bulls also had a higher intake of the intensive diet….not only show the scientific results, but also economic results for the producers (the authors show advantages from the consumer perspective but not higher costs for the producers). Please include some comments on this in the discussion.

R: We add a sentence “In Poland 90% of beef is produced in dairy herds and therefore the application of the optimal for Polish Holstein-Friesian bulls feeding technology with finishing phase might improve beef quality [5].” To show that in Poland the strategy is not used and therefore might be regarded as a novelty. The modification in the discussion section were made, with a notification of increased cost of beef production in the intensive system “It should be highlighted that feed costs constitute the main part of the total costs of beef production, regardless of the fattening system adopted [Jennings et al. 2018, 41]. In our research, the intensive system was associated with greater consumption of concentrate feed. Therefore, the effect of increased slaughter value obtained in the intensive system should be compared with the increased costs of feeding these bulls.”

Specific comments:

C2. L 49-50….large and late maturating animals whose carcass quality improves along with body weight increase [6]. In contrast to small and early maturating cattle…... I am not sure if the correct word is late maturating or late maturing breeds, please check English wording

R. Thank you, it should be “maturing”. The issue was corrected in the text.

C3. L169…were used to fabricate 2.54 cm-thick steaks. I suggest to use cut instead of fabricate

R. corrected

C4. Table 5…the SEM of cooking loss (%), 2.295 should be reduced to 2 decimals, as the rest of values

R. corrected

Reviewer 2 Report

General comments

The manuscript is a topical issue. The context in which the study was conducted was relevant thus making the work important for beef production. The paper was mainly carefully prepared. The manuscript needs moderate revisions, particularly when discussing the results and making conclusions. English and particularly the terminology should be improved.

 Please find the comments below and take them as constructive criticisms for this manuscript.

Title

In the title finish fattening score is unclear. In my opinion, fattening score includes in carcass value.

Suggestion for the title: Effects of silage type and feeding intensity on carcass traits and meat quality of finishing Holstein-Friesian Bulls

 Simple summary

Line 17: Prefer feeds than fodder. The same comment throughout the whole paper. Lines 18: Prefer ´carcass and meat quality´ than ´fattening progress´

Line 18: Prefer level than constituent

Line 18: Consider to remove ´beneficially´

Line 19: Consider to remove ´fattening progress´ or replace it with ´performance´

Line 19: Consider ´The intensive feeding improved performance, carcass quality and meat quality of HF bulls.´

 Abstract

Lines 20-23: The sentence should be clarified.

 Introduction

Lines 41-44: The sentence “The main factors…” could be placed before the sentence “The beef originating from bulls…”.

Lines 42-43: The main factors are just listed. The influence (increases or decreases IMF) of these factors could be shortly mentioned.

Line 45: …which makes them available…

Lines 52-53: Does that mean HF bulls? If yes, the breed could be mentioned.

Line 53: Should here be concentrate rather than concentrated feed?

Line 54: ad libitum

Line 55: Bulk feed is unclear. It should be clarified what it means.

Line 58: Consider to use some other term for ´application´. Maybe feeding system or feeding method

Lines 58-59: “Finishing phase” should be defined (energy-dense diet, typically increased concentrate proportion in the diet) before its effects on carcass and meat quality. Consider also could “ending the fattening” be removed because “finishing phase” as such is commonly used.

Line 61: There is a contradiction between sentence in lines 50-51 ´HF is suitable for long fattening, which does not lead to excessive fat tissue deposition´ and sentence in line 61 ´improve fat deposition resulting in benefits´ and this should be taken into account somehow. Which one is the target, improved fat tissue deposition or not? In addition, for the meat industry, high-fat carcasses may cause additional expenditures. But also effects on meat quality are important to be taken into account.

Lines 74-75: This sentence should be clarified.

Lines 78-79: ´…energy level concentration in a fodder portion on finishing fattening results, carcass value and meat quality of Holstein-Friesian young bulls was undertaken.´ could be replaced with ´…feeding intensity during finishing phase on performance, carcass value and meat quality of Holstein-Friesian bulls.´

Lines 80-84: Suggestion for modification the beginning and the end of the sentence: A hypothesis that maize silage supplemented with a high proportion of concentrate has… than grass silage supplemented with a lower proportion of concentrate.

Lines 80-84: Hypotheses should be more precise and detailed.

Materials and Methods

Line 88: The experiment comprised…

Line 93: How many animals per group?

Lines 93-94: First the whole word and then the abbreviation (TMR) in brackets

Line 98: Consider to end the sentence after the word ´each´ and then start a new sentence ´They were placed…´

Line 100: m2 per bull could be mentioned

Line 100: Consider to remove ´had enough space to move freely around´

Line 102: Prefer ´During two weeks pre-experimental period the bulls were adapted to housing conditions and feeding. After this the feeding experiment started. ´

It also should be reported here the age and weight of the bulls at the start of the feeding experiment.

Line 103: Consider to remove ´fattening´ throughout the whole paper in the corresponding points

Line 103: …were fed with two different types of silage and supplemented with two different levels of concentrate…

Line 104: Grass species and varieties and variety of maize should be added. Also, additive should be mentioned if it was used.  

Line 106: Information of triticale should be added, including variety, harvesting, storage, processing before feeding…

Table 1. Consider to remove (g/kg dry matter, DM) from the title and add it into the table where it is needed.

Line 110: Explanations for abbreviations I and SI are needed.

Lines 114-115: Remove (Total Mixed Ration)

Line 115: What was the content of the premix? If this means the same than commercial mineral-vitamin premix, clarify in the text.

Line 120: If the animals were weighed at the beginning and at the end of the experiment it should be also mentioned. And if these were conducted on two consecutive days it should be mentioned?

line: Replace had been with were

Line 121: The daily feed intake

Line 124: Prefer analysed than assayed, the same comment throughout the whole paper in the corresponding points.

Table 2. Prefer proportions in g/kg DM than %

Line 143: In this chapter should be presented how carcass weight was calculated

Lines 144-145: Maybe prefer slaughterhouse

Line 145: Prefer pen than box

Lines 145-146: ´ The animals were weighed before and after slaughter with an accuracy of 0.5 kg.´ This sentence should be clarified taken into account that before the slaughter it was live weight and after the slaughter it was carcass weight.

Line 150: The value of meat pH…

Line 156: Meat sampling

Lines 157-158: The cutting point should be indicated more precisely

Line 157: For meat quality analyses longissimus lumborum (LL) was removed from…

Lines 157-158: Add the size (g) of the sample

Line 158: Prefer The LL or sample than muscles. The same comment throughout the whole paper in the corresponding points.

Line 161-162: …individually vacuum packed (PA/PE…

Lines 164-165: The sentence should be clarified

Line 165: After XX days ageing, the samples were put in the freezer…

Line 167: Prefer sample than portion

Line 168: Clarify what is included in the chemical composition

Line 169: Start a new sentence ´A 200 g sample was cut and used…

Line 169: Fabricate is unclear and the end part of the sentence should be clarified. Also mention here how many steaks per one sample and for what analyses the steaks were cut.

Line 173: Was one steak used for both sensory quality and cooking loss?

Line 176: remove A portion of and start with For the chemical composition analyses approx. 200 g of LL…

Line 175: Specify what was analysed

Line 182: Specify which value was used as a result

Line 202: If correct, add ´meat colour was measured from a freshly cut surface´

Line 205: Specify which value was used as a result

Line 216: Specify which value was used as a result

Line 240: Were live weight and carcass weight included in the statistical model for assessing their possible effects?

Material and methods: In this chapter should be presented how live weight gain was calculated. Add to this chapter what the slaughter date was based on. Age or carcass weight or the length of the growing period? Also, what was the target age, weight or length.

Results

General comment related to the Tables that SEM should be presented with one more decimal than that of the result of the parameter.

Line 258: Prefer live weight than body weight. The same comment throughout the whole paper in the corresponding points.

Lines 260-261: Replace ´A higher growth rate of the maize silage-fed bulls was not proved statistically.´ with Silage type did not affect growth rate.

Line 264: The feed intake of the bulls in intensive diet was higher (P<0.05) and the feed conversion ratio was lower (add P-value) compared to semi-intensive diet.

Figure 1: The scale should start from zero

Line 270: Consider ´Carcass quality´

Line 270: Carcass weight and dressing proportion should also be presented here.

Line 273: Clarify compared to what. Body weight should be presented in Fattening result.

Lines 273-274: Consider to remove ´Carcass weight was also affected (P<0.01) by feeding intensity –´

Line 275: Clarify compared to what.

Line 276: Consider to remove ´ The feeding intensity differentiated also carcass conformation and fatness.´

277-278: Prefer: Conformation and fat scores of intensively fed bulls were higher (P<0.01) than semi-intensively fed bulls.

Table 4. Include pH48 prefer into the meat quality than slaughter traits

Table 4. Is there some reason for *?

Table 5: Prefer proportions in g/kg than %

Discussion

As a general comment English should be improved with a special attention to the terminology. Hypotheses should be taken into consideration more clearly.

Line 323: More than one reference should be included

Line 322-323: Effect of energy intake on growth rate could be discussed more and maybe use as one reference Huuskonen, A. & Huhtanen, P. 2015. The development of a model to predict BW gain of growing cattle fed grass silage-based diets. Animal 9, 1329–1340. https://doi.org/10.1017/S1751731115000610.

Lines 323-325: This sentence should be clarified. Also, experimental feed compositions and feed values particularly the small difference in UFV, should be taken into account and discuss their possible effects on the results.

Lines 331-334: Importance of energy intake and effect on pH could also be mentioned.

Lines 387: Maybe here could be discussed that fat contents were very low and the difference between I and SI were rather small, and this may be one reason that no differences in WBSF were found.

Line 390: Effects on fatty acids is loose here and is not related to WBSF which is discussed here.

Conclusions

Generally, conclusions are now more results than conclusion. This chapter should be improved, and main conclusions should be more highlighted.

Line 405: Highlight prefer energy intake than fibre intake

English and particularly the terminology should be improved throughout the paper.

Author Response

Dear Reviewer, thank you very much for your detailed comments, which enables us to improve the manuscript. Below we enclosed responses for each comment. In the manuscript, all changes were marked with red font.

 C1. General comments. The manuscript is a topical issue. The context in which the study was conducted was relevant thus making the work important for beef production. The paper was mainly carefully prepared. The manuscript needs moderate revisions, particularly when discussing the results and making conclusions. English and particularly the terminology should be improved. Please find the comments below and take them as constructive criticisms for this manuscript.

Thank you very much for your detailed comments. We truly appreciate your work and commitment. We did our best to improve the quality of our manuscript following your comments.

C2. Title. In the title finish fattening score is unclear. In my opinion, fattening score includes in carcass value. Suggestion for the title: Effects of silage type and feeding intensity on carcass traits and meat quality of finishing Holstein-Friesian Bulls

Thank you for the comment. The title was changed following your suggestion.

Simple summary.

C3. Line 17: Prefer feeds than fodder. The same comment throughout the whole paper.

The word “fodder” was replaced by “feed” throughout the manuscript.

C4. Lines 18: Prefer ´carcass and meat quality´ than ´fattening progress´

Corrected

C5. Line 18: Prefer level than constituent

We changed it for “the concentration of energy” expression

C6. Line 18: Consider to remove ´beneficially´

We would rather keep the word “beneficially” since it is a simple summary and to give readers a take-home message easy to understand.

C7. Line 19: Consider to remove ´fattening progress´ or replace it with ´performance´

Deleted

C8. Line 19: Consider ´The intensive feeding improved performance, carcass quality and meat quality of HF bulls.´

Corrected following your comment.

 Abstract

C9. Lines 20-23: The sentence should be clarified.

We rephrased the sentence to make it more easy to follow.

 Introduction

C10. Lines 41-44: The sentence “The main factors…” could be placed before the sentence “The beef originating from bulls…”.

Changed

C11. Lines 42-43: The main factors are just listed. The influence (increases or decreases IMF) of these factors could be shortly mentioned.

We added more text, explaining the impact of mentioned factors on IMF in beef.

C12. Line 45: …which makes them available…

Corrected

C13. Lines 52-53: Does that mean HF bulls? If yes, the breed could be mentioned.

Yes, the breed name was added.

C14. Line 53: Should here be concentrate rather than concentrated feed?

Corrected in the whole manuscript

C15. Line 54: ad libitum

Corrected

C16. Line 55: Bulk feed is unclear. It should be clarified what it means.

The expression “bulk feed” was replaced by “roughage”

C17. Line 58: Consider to use some other term for ´application´. Maybe feeding system or feeding method

Thank you for the suggestion. The sentence was rewritten.

C18. Lines 58-59: “Finishing phase” should be defined (energy-dense diet, typically increased concentrate proportion in the diet) before its effects on carcass and meat quality. Consider also could “ending the fattening” be removed because “finishing phase” as such is commonly used.

Thank you for the suggestion. The sentence was rewritten. The information about the finishing phase added following your comment.

C19. Line 61: There is a contradiction between sentence in lines 50-51 ´HF is suitable for long fattening, which does not lead to excessive fat tissue deposition´ and sentence in line 61 ´improve fat deposition resulting in benefits´ and this should be taken into account somehow. Which one is the target, improved fat tissue deposition or not? In addition, for the meat industry, high-fat carcasses may cause additional expenditures. But also effects on meat quality are important to be taken into account.

Thank you for the comment. The sentences were revised and corrected. We specify what kind of fat we bear in mind: subcutaneous or IMF.

C20. Lines 74-75: This sentence should be clarified.

The sentence was modified. We hope it is more clear now.

C21. Lines 78-79: ´…energy level concentration in a fodder portion on finishing fattening results, carcass value and meat quality of Holstein-Friesian young bulls was undertaken.´ could be replaced with ´…feeding intensity during finishing phase on performance, carcass value and meat quality of Holstein-Friesian bulls.´

The sentence was modified according to your suggestion.

C22. Lines 80-84: Suggestion for modification the beginning and the end of the sentence: A hypothesis that maize silage supplemented with a high proportion of concentrate has… than grass silage supplemented with a lower proportion of concentrate. And C 23. Lines 80-84: Hypotheses should be more precise and detailed.

Thank you. The hypothesis was revised and rewritten.

Materials and Methods

C24. Line 88: The experiment comprised…

Corrected

C25. Line 93: How many animals per group?

The were 8 animals per group. The information is provided in a paragraph were the experimental groups were described (next paragraph). At the beginning the animals were not divided.

C26. Lines 93-94: First the whole word and then the abbreviation (TMR) in brackets

Corrected

C27. Line 98: Consider to end the sentence after the word ´each´ and then start a new sentence ´They were placed…´

Modified

C28. Line 100: m2 per bull could be mentioned

Added

C29. Line 100: Consider to remove ´had enough space to move freely around´

We deleted the information.

C30. Line 102: Prefer ´During two weeks pre-experimental period the bulls were adapted to housing conditions and feeding. After this the feeding experiment started. ´

Modified following your comment.

C31. It also should be reported here the age and weight of the bulls at the start of the feeding experiment.

The information was given at the beginning of the paragraph and in the results section (Table 3) and discussed.

C32. Line 103: Consider to remove ´fattening´ throughout the whole paper in the corresponding points

We revised the manuscript and replace “fattening” with feeding or finishing phase where appropriate. The word

C33. Line 103: …were fed with two different types of silage and supplemented with two different levels of concentrate…

Corrected

C34. Line 104: Grass species and varieties and variety of maize should be added. Also, additive should be mentioned if it was used. 

The information was added.

C35. Line 106: Information of triticale should be added, including variety, harvesting, storage, processing before feeding…

The information was added.

C36. Table 1. Consider to remove (g/kg dry matter, DM) from the title and add it into the table where it is needed.

Modified according to your comment

C37. Line 110: Explanations for abbreviations I and SI are needed.

Explanations were added.

C38. Lines 114-115: Remove (Total Mixed Ration)

Removed

C39. Line 115: What was the content of the premix? If this means the same than commercial mineral-vitamin premix, clarify in the text.

Yes, it was the same commercial premix.

C40. Line 120: If the animals were weighed at the beginning and at the end of the experiment it should be also mentioned. And if these were conducted on two consecutive days it should be mentioned?

The animals were weighed at the beginning of the experiment and then once a week. To the sentence mentioned by you, we added information about weighing them at the beginning of the experiment. They were also weighed before and after slaughter, and the information is provided in a more suitable place at the end of the section.

C41. line: Replace had been with were

Corrected

C42. Line 121: The daily feed intake

Corrected

C43. Line 124: Prefer analysed than assayed, the same comment throughout the whole paper in the corresponding points.

Corrected

C44. Table 2. Prefer proportions in g/kg DM than %

We prefer to keep data in % because it presents differences related to the "feeding intensity" factor better than g/kg.

C45. Line 143: In this chapter should be presented how carcass weight was calculated

The carcasses were weighed. The information on how the dressing percentage was calculated was provided in 2.2. section.

C46. Lines 144-145: Maybe prefer slaughterhouse

Corrected

C47. Line 145: Prefer pen than box

Corrected

C48. Lines 145-146: ´ The animals were weighed before and after slaughter with an accuracy of 0.5 kg.´ This sentence should be clarified taken into account that before the slaughter it was live weight and after the slaughter it was carcass weight.

The sentence was modify to make it more clear “The animals were weighed before slaughter as well as their carcasses with an accuracy of 0.5 kg and reported as body weight before slaughter and hot carcass weight, respectively.”

C49. Line 150: The value of meat pH…

Corrected

C50. Line 156: Meat sampling

Corrected

C51. Lines 157-158: The cutting point should be indicated more precisely

The information was added.

C52. Line 157: For meat quality analyses longissimus lumborum (LL) was removed from…

Modified

C53. Lines 157-158: Add the size (g) of the sample

The sample size is given in the further sentence – “The next day, the samples (approx. 700 g) were individually….”

C54. Line 158: Prefer The LL or sample than muscles. The same comment throughout the whole paper in the corresponding points.

Modified in the whole manuscript where applicable

C55. Line 161-162: …individually vacuum packed (PA/PE…

Modified according to the comment

C56. Lines 164-165: The sentence should be clarified

The sentence was modified to make it more clear

C57. Line 165: After XX days ageing, the samples were put in the freezer…

Modified according to the comment

C58. Line 167: Prefer sample than portion

Changed

C59. Line 168: Clarify what is included in the chemical composition

The information was added

C60. Line 169: Start a new sentence ´A 200 g sample was cut and used…

Modified

C61. Line 169: Fabricate is unclear and the end part of the sentence should be clarified. Also mention here how many steaks per one sample and for what analyses the steaks were cut.

The sentence was modified to make it more clear.

C62. Line 173: Was one steak used for both sensory quality and cooking loss?

Yes, it was the same steak.

C63. Line 176: remove A portion of and start with For the chemical composition analyses approx. 200 g of LL…

Modified according to the comment

C64. Line 175: Specify what was analysed

Specified

C65. Line 182: Specify which value was used as a result

The value of pH was measured in triplicate in each sample and mean values for each group were reported in a table. In the manuscript the information was included.

C66. Line 202: If correct, add ´meat colour was measured from a freshly cut surface´

The information was added.

C67. Line 205: Specify which value was used as a result

The colour was measured in triplicate in each sample and mean values for each group were reported in a table. In the manuscript the information was included.

C68. Line 216: Specify which value was used as a result

The shear value was measured in 5 repetitions in each sample and mean values for each group were reported in a table. In the manuscript the information was included.

C69. Line 240: Were live weight and carcass weight included in the statistical model for assessing their possible effects?

No.

C70. Material and methods: In this chapter should be presented how live weight gain was calculated. Add to this chapter what the slaughter date was based on. Age or carcass weight or the length of the growing period? Also, what was the target age, weight or length.

The information was provided: “Finishing fattening was carried out for 120 days. Body weight gains during finishing fattening were calculated by subtracting the initial weight from the final weight, and the resulting difference was divided by 120.”

Results

C71. General comment related to the Tables that SEM should be presented with one more decimal than that of the result of the parameter.

Corrected according to the comment

C72. Line 258: Prefer live weight than body weight. The same comment throughout the whole paper in the corresponding points.

Corrected in the whole manuscript

C73. Lines 260-261: Replace ´A higher growth rate of the maize silage-fed bulls was not proved statistically.´ with Silage type did not affect growth rate.

The sentence was replaced

C74. Line 264: The feed intake of the bulls in intensive diet was higher (P<0.05) and the feed conversion ratio was lower (add P-value) compared to semi-intensive diet.

The sentence was replaced

C75. Figure 1: The scale should start from zero

Corrected

C76. Line 270: Consider ´Carcass quality´

Changed for carcass quality

C78. Line 270: Carcass weight and dressing proportion should also be presented here.

We changed the subtitle to “carcass quality” and believe that it cover carcass weight and dressing proportion.

C79. Line 273: Clarify compared to what. Body weight should be presented in Fattening result.

The sentence was clarified. Body weight was presented in a section 3.1. Feeding results (line 269). In this section we only show the link between those parameters, which means that it wasn’t surprising.

C80. Lines 273-274: Consider to remove ´Carcass weight was also affected (P<0.01) by feeding intensity –´

Removed

C81. Line 275: Clarify compared to what.

It was clarified

C82. Line 276: Consider to remove ´ The feeding intensity differentiated also carcass conformation and fatness.

Removed

C83. 277-278: Prefer: Conformation and fat scores of intensively fed bulls were higher (P<0.01) than semi-intensively fed bulls.

Changed as suggested

C84. Table 4. Include pH48 prefer into the meat quality than slaughter traits

The pH 48h results were moved to Table 4.

C85. Table 4. Is there some reason for *?

No, there is no need for that. We deleted asterisks.

C86. Table 5: Prefer proportions in g/kg than %

The chemical composition of food products, including meat, is typically presented in %. Therefore we would like to keep %.

Discussion

C87. As a general comment English should be improved with a special attention to the terminology. Hypotheses should be taken into consideration more clearly.

Terminology was improved.

C88. Line 323: More than one reference should be included

We cited more references

C89. Line 322-323: Effect of energy intake on growth rate could be discussed more and maybe use as one reference Huuskonen, A. & Huhtanen, P. 2015. The development of a model to predict BW gain of growing cattle fed grass silage-based diets. Animal 9, 1329–1340. https://doi.org/10.1017/S1751731115000610.

Thank you for the suggestion. We added that reference

C90. Lines 323-325: This sentence should be clarified. Also, experimental feed compositions and feed values particularly the small difference in UFV, should be taken into account and discuss their possible effects on the results.

The paragraph was expanded following your comment. Thank you.

C 91. Lines 331-334: Importance of energy intake and effect on pH could also be mentioned.

The importance of feeding intensity on meat’s pH were mentioned in line 346. The importance of energy intake was discussed following your suggestion.

C92. Lines 387: Maybe here could be discussed that fat contents were very low and the difference between I and SI were rather small, and this may be one reason that no differences in WBSF were found.

Thank you for the comment. It was discussed earlier in lines 370-372.

C93. Line 390: Effects on fatty acids is loose here and is not related to WBSF which is discussed here.

The information was deleted.

Conclusions

C94.Generally, conclusions are now more results than conclusion. This chapter should be improved, and main conclusions should be more highlighted. And C95. Line 405: Highlight prefer energy intake than fibre intake

We revise conclusions section and made correction to highlight the beneficial effect of energy intake.

C96. Comments on the Quality of English Language. English and particularly the terminology should be improved throughout the paper.

We would like to thank you for your numerous comments about English terminology. We did our best to improve the manuscript. Your comments were greatly helpful for us in revision of the paper. Thank you for your time and commitment.

Round 2

Reviewer 2 Report

General comments

The revised manuscript has been improved and clarified. However, I still have some comments and some revisions are needed. Please, find comments below.

Abbreviations should be used systematically. First time write the whole word and add abbreviation in the brackets, and then use only abbreviation. Many abbreviations are too late, or they are not used systematically. For example, Holstein-Friesian, grass silage, maize silage, longissimus lumborum… Revise these throughout the manuscript.

Introduction

Line 57: …in which silage feeding ad libitum…

Line 66: Maybe prefer method than technology

Materials and Methods

Line 160: …individual pens with…

Lines 202, 225, 240: Reference to the table is not needed in Materials and methods and could be removed. If reference to the Table remains, number of the Table should be added.

Lines 233-234: The sentence should be clarified.

Author Response

Dear Reviver, we would like to express our gratitude for your valuable comments which enables us to improve the paper. We truly appreciate your work and commitment. Thank you! We revised the manuscript carefully and responded point-by-point for Reviewer's comments. The manuscript was prepared in “track changes” mode, however to ease the reading and show the changes, red font was also used to indicate changed words. To make corrections we used the version of the manuscript found at MDPI system

General comments. The revised manuscript has been improved and clarified. However, I still have some comments and some revisions are needed. Please, find comments below.

Comment 1: Abbreviations should be used systematically. First time write the whole word and add abbreviation in the brackets, and then use only abbreviation. Many abbreviations are too late, or they are not used systematically. For example, Holstein-Friesian, grass silage, maize silage, longissimus lumborum… Revise these throughout the manuscript.

Response: The manuscript was revised and abbreviations and their usage corrected throughout the manuscript.

Introduction

Comment 2: Line 57: …in which silage feeding ad libitum…

Response: Corrected

Comment 3: Line 66: Maybe prefer method than technology

Response: Corrected

Materials and Methods

Comment 4: Line 160: …individual pens with…

Response: Corrected

Comment 5: Lines 202, 225, 240: Reference to the table is not needed in Materials and methods and could be removed. If reference to the Table remains, number of the Table should be added.

Response: The reference to a table was deleted

Comment 6: Lines 233-234: The sentence should be clarified.

Response: We rewritten the sentence in attempt to make it more clear.